# Gamma, E-Beam and X-ray Irradiations on PE/EVOH/PE Multilayer Film: An Industrial Point of View Regarding the Impact on Mechanical Properties

**DOI:** 10.3390/polym15132799

**Published:** 2023-06-24

**Authors:** Nina Girard-Perier, Sylvain R. A. Marque, Nathalie Dupuy, Blanche Krieguer, Samuel Dorey

**Affiliations:** 1Sartorius Stedim FMT S.A.S, Z.I. Les Paluds, Avenue de Jouques CS91051, CEDEX, 13781 Aubagne, France; blanche.krieguer@sartorius.com; 2Aix Marseille Univ, CNRS, ICR, Case 551, 13397 Marseille, France; sylvain.marque@univ-amu.fr; 3Aix Marseille Univ, Avignon Université, CNRS, IRD, IMBE, 13013 Marseille, France; nathalie.dupuy@univ-amu.fr

**Keywords:** irradiation technology, gamma, X-rays, polyethylene, mechanical properties, thermal and viscoelastic properties

## Abstract

X-ray and electron-beam (E-beam) sterilization technologies were assessed to supplement gamma sterilization, the most common radiation technology used today for biopharmaceutical product sterilization. The mechanical properties of a PE/EVOH/PE film were studied using tensile tests and dynamical mechanical analysis after each irradiation technology (i.e., gamma, electron beam and X-ray irradiations). The effects of each irradiation were compared using two statistical methods. The results indicate that the three irradiation technologies induce no difference in mechanical properties in the investigated dose range for this material.

## 1. Introduction

Single-use plastic bags are frequently used by the biotechnological and biopharmaceutical industries for the storage, shipping or preparation of intermediates, biopharmaceutical preparations and solutions and final products [1,2,3]. Films used in single-use systems are mainly made, for example, of polyethylene/ethylene vinyl alcohol/polyethylene (PE/EVOH/PE) multilayer film. These polymers are selected for their properties of barrier protection and flexibility to ensure the integrity of the solution. Single-use plastic bags in biopharmaceutical and biotechnological environments require sterilization to be free of microorganisms [4]. The dose range usually used for biopharmaceutical industries is 25–45 kGy [5]. Nowadays, the most common radiation technology for pharmaceutical devices is gamma radiation. The current healthcare context leads to a strong growth in the manufacture of biopharmaceutical products. In addition, there are concerns about the capacity of sterilization using gamma radiation in future years [6]. Two alternative methods, electron beam irradiation and X-ray irradiation, are being considered for use as additional technologies to gamma sterilization.

Electron beam irradiation is generated when electrons are accelerated by an electromagnetic field in an accelerator. X-rays are photons that are generated using an electron beam accelerator. The electron beam is converted to photons by accelerating the electrons into a high-density material (tungsten, tantalum) with an energy in the same range as gamma. It is important to investigate the impact of these two additional sources on single-use systems. For both irradiation modalities, a sterility assurance level (SAL) of 10^−6^ can be achieved. The ionizing radiation energy for the gamma, E-beam and X-ray radiations considered is used to define the absorbed dose (in kGy) because it is the ionizing ability of the radiation—the ability to ionize or kick electrons off the shell of atoms through Compton scattering—that initiates the killing effect on microorganisms. Up to now, the impact of gamma irradiation on both EVOH and PE polymers has been exhaustively studied [7,8,9,10,11,12,13,14,15,16,17,18,19,20,21]. Few studies report on the impact of the two other irradiation technologies (i.e., E-beam and X-rays) [22,23,24,25]. Two studies report on the effect of gamma irradiation and E-beam irradiation on polypropylene [26,27]. The authors observed that gamma irradiation has a greater effect on physical, mechanical and thermal properties than E-beam irradiation. Badia et al. [28] compared gamma and electron beam irradiation on high-density polyethylene shelf life and found no change. Croonenborghs et al. [29] observed a similar effect between gamma and X-ray irradiations on the mechanical properties of polyethylene.

Regarding the PE/EVOH/PE multilayer film in this paper, many studies were performed on the impact of gamma irradiation using various analytical techniques and at different analysis levels, such as molecular level with electron spin resonance [30] (to monitor radicals) or High-Pressure Liquid Chromatography [31] (to monitor oxidations), analysis at macromolecular level with X-ray Photoelectron Spectroscopy [32] or analysis at material level with permeation or mechanical properties [33]. However, the lack of data in the literature on the impact of electron beam and X-ray irradiations on the mechanical properties of multilayer materials could hamper their use. 

The use of ionizing irradiation as a sterilizing method implies the occurrence of two mains reactions: bond cleavage and cross-linking—which may alter the physical and mechanical properties of the polymers. The cross-linking and chain scission events related to the effect of radiation on molecular weight distribution are well described in the literature [34,35].

For characterization and qualification studies, equivalence between process parameter outputs needs to be shown. The impact of three irradiation technologies, X-ray, E-beam and gamma irradiations is investigated in industrial conditions at a single dose of 50 kGy on single-use bags made up of PE/EVOH/PE multilayer film. This approach has a strong limitation due to the non-controlled environment, which was circumvented by paying attention to the repeatability and reproducibility of the experiments. Typically, hypothesis testing is used as a data analysis technique to evaluate the comparability of data. Our approach here aims to be as close as possible to the industrial conditions of irradiations, and two datasets are comparable if their difference in means is of no practical and statistical significance such that it is accepted as nearly zero [36]. Today, a simple equivalence test first introduced by Schuirmann et al. [37] is accepted, for example, as standard for bioequivalence assessment [38], namely, the two one-sided *t*-test (TOST). In contrast to the two-sample *t*-test, the null hypothesis of the TOST states that the two means are not equivalent [36]. The impact of the null hypothesis is that in the case of small sample sizes and/or poor precision (large variance) in one or both groups, equivalence is rejected, resulting in low numbers of false-positive test results [39]. With such test results, conclusions can also be difficult to draw from very large sample sizes. A very large sample size in the reference data leads to a narrow equivalency acceptance criterion (EAC), and therefore, equivalence is easily rejected even though the groups are practically equivalent [40]. Modifications induced upon irradiation were monitored, and the potential differences induced between irradiation technologies were investigated with the Student test (i.e., *t*-test) and the two one-sided test (TOST) equivalence test. In this paper all these statistical tests are performed on mechanical performances [1].

## 2. Materials and Methods

### 2.1. PE/EVOH/PE Multilayer Film

The structure of the PE/EVOH/PE multilayer film analyzed in this article is depicted in Figure 1. It has a total thickness of 400 µm.

### 2.2. Storage Conditions

Before irradiation, each sample was wrapped in a packaging bag (PE/Polyamide/PE), and a partial vacuum (not controlled) was formed [27]. However, materials were not O_2_-degassed and therefore contained some level of O_2_. The packaging bags were placed in a cardboard box and then irradiated. After irradiation and before analysis, the cardboard box was stored in the dark, in an air-conditioned room at 20 ± 2 °C. The details of storage time (ageing) after irradiation are shown in Table 1.

### 2.3. Irradiation Methods

Gamma

PE/EVOH/PE multilayer films were irradiated at room temperature with a ^60^Co gamma source at BGS—Beta Gamma Service GmbH & Co. KG, Wiehl, Germany. The dose rate provided was 1–2 kGy/h. The irradiation was performed at room temperature under an environmental atmosphere. The detail of doses received for each sample depending on analysis is reported in Table 1.

Electron beam

PE/EVOH/PE multilayer films were packed and wrapped in a packaging bag (PE/Polyamide/PE) and placed side by side in an 8 cm thin cardboard box to have the same thickness of plastic material. They were irradiated with a 10 MeV Rhodotron (Ionisos, Chaumesnil, France), with a power source at 28 kW. The dose rate was 300 kGy/min. Alanine dosimeters were used on the cardboard box to assess the radiation delivered to the single-use bag samples (± 5%). The doses received for each sample depending on analysis are reported in Table 1.

X-ray

PE/EVOH/PE multilayer films were irradiated with a 7 MeV Rhodotron (Steris, Däniken, Switzerland), with a power source at 360 kW. The average dose rate was 80 kGy/h. The doses received for each sample depending on analysis are reported in Table 1.

### 2.4. Tensile Strength Measurement

Yields, ultimate tensile strength (UTS) and elongation were measured, with specimens along the machine direction (MD) and transverse direction (TD). The MD corresponds to the direction the film was extruded (main direction of the polymer chains), while TD is perpendicular to MD. 

The specimens were cut according to the ISO527-3 [41] (150 mm length, 15 mm width). The crosshead speed used was 500 mm/min. The initial gauge length was 100 mm. Six to eight specimens per direction were used for each sample. 

The tensile strengths of irradiated PE/EVOH/PE multilayer films were measured using a universal testing machine (Zwick Z005) at 23 ± 3 °C. UTS is the maximum tensile load a material can withstand prior to fracture and is calculated by dividing the maximum load by the original cross-sectional area of the specimen. All raw data can be found in the Appendix A.

### 2.5. DMA (Dynamic Mechanical Analysis)

DMA was performed using the DMA 850 Discovery model from TA instruments. Tests were performed according to ASTM 1640-04. The storage (elasticity) modulus (E′), the loss (viscosity) modulus (E″) and the tangent delta (tan δ, ratio of the loss to the storage curves) were studied as a function of temperature. The storage modulus is a measure of the elastic character of the material [42]. The loss modulus is a measure of the viscous character of the material [42]. The tan δ is defined as the ratio of the loss modulus to the storage modulus (i.e., $\tan\delta =\frac{E^{\prime \prime }}{E}$). It represents the ratio of energy dissipated to energy stored per cycle of deformation [42]. We considered maxima in curves to assess the different thermal transitions. The γ transition (or glass transition) of the PE of the PE/EVOH/PE multilayer film is obtained with the maximum of the tan δ curve. The β transition is defined as the maximum of the loss modulus curve. The α transition was not studied as there are some variations due to a specimen slipping effect from approx. 50 °C and the overlapping of the glass transition of EVOH and the α transition temperature of PE.

Measurements were performed with 1 Hz by increasing the temperature from −150 °C to 110 °C. Between 2 and 6 specimens were tested per irradiation technology; the details and all the raw data are in the supporting information file (Appendix A).

### 2.6. Statistical Analyses

#### 2.6.1. Equivalency Assessment

The statistical evaluation was conducted with a two one-sided test (TOST) equivalence test using the software Minitab^®^. The equivalency acceptance criterion (EAC) is used to check if the measurement results obtained from one type of irradiation to another fall within the equivalence interval. It is defined as the limit outside which the difference in mean values should be considered practically and statistically significant [39].

#### 2.6.2. Tensile Strength EAC

The first parameter that must be specified before an analyst performs statistical testing is δ, i.e., the equivalence acceptance criterion (EAC), the absolute value of the true difference between the groups’ mean values. δ is a hypothetical value such that if the absolute value of the observed difference is no more than δ, there is a strong probability of concluding that the two datasets represent equivalent results [40]. There are different methods to calculate the equivalence acceptance criterion [36], such as the 3-sigma rule (the one used in this paper), Limentani’s EAC, Limentani modified, Cohen’s standardized difference—effect sizes—and tolerance interval.

To allow the acceptance of a measurable impact on the polymer, criterion δ needs to consider the intrinsic variabilities of the materials and variabilities from the methodology. These variabilities were studied on UTS (Ultimate Tensile Stress) and elongation. The output summary is available in Table 2. These variations reproduce the potential variations during a comparative study over time. The different factors involved in characterizing the materials and method measurements are close to potential variations met during a comparative study, meaning at least different sampling in one batch (destructive tests) and two operators.

The equivalence criteria δ are drawn to exceed the six-sigma variations of the materials and the methodology variabilities to significantly discriminate and to conclude on the impact of the studied factor. Therefore, for the UTS, to conclude on a potential impact, the criterion selected is 35% to also embed other films to assess (reproducibility not shown here). The ratio (UTS_X-rays_/UTS_gamma_) must be between 0.65 and 1.35 to claim equivalency. For the elongation, the criterion of 25% is selected to conclude on a potential impact on the materials in accordance with other studies [43]. The ratio (Elongation_X-rays_/Elongation_gamma_) must be between 0.75 and 1.25. The six-sigma variation for films was evaluated in the transversal direction. The criteria determined are also applicable for the machine direction. As a reminder, in TOST, *p*-value_TOST_ must be <0.05 with α = 5% to claim equivalency, while with the *t*-test, two means are not considered different when *p*-value*_t_*_-test_ > 0.05.

#### 2.6.3. Thermal Properties EAC

As DMA analysis allows us to study the thermal properties as Differential Scanning Calorimetry (DSC) does, we selected the same criterion for both experiments. With repetitive DSC measurements on one batch of several materials, two sample picking and by two operators, a variability of 5 °C was defined as the DSC criterion. A temperature of 5 °C was then also chosen for DMA analysis to evaluate a potential impact on the material. The statistical evaluation was performed on thermal transitions observed in the storage modulus (E′), loss modulus (E″) and tangent delta (tan δ) curves. 

The null hypothesis H0 in TOST is either H0_1_ = difference ≤ −5 °C or H0_2_ = difference ≥ 5 °C. When both these one-sided tests can be statistically rejected, we can conclude that the alternative hypothesis H1 = −5 °C < difference < 5 °C is true and that the observed effect falls within the equivalence bounds and is crossing or touching zero and is, therefore, practically equivalent (or, in other words, statistically not different).

## 3. Results

### 3.1. Tensile Strength Evaluation

An overview of the tensile curves of the PE/EVOH/PE film obtained after each irradiation technology (i.e., gamma in red, X-rays in blue and electron beam in green) is plotted in Figure 2.

For the non-sterile and sterile samples, four zones can be defined. Two yields, at ≈15% and ≈50%, are observed as zones I and II. From ≈50% elongation, there is a plateau to ≈200% elongation for zone III. From ≈200% (zone IV), strain hardening starts until the complete rupture of the film up to ≈500%. The UTS results obtained for the irradiated multilayer film are depicted in Figure 3a for MD and Figure 3b for TD. Each tensile curve was decomposed into six characteristic values: ultimate tensile strength at break (UTS), ultimate elongation or elongation at break, 1st Yield Strength (Y1 Strength), 1st Yield Strain (Y1 Strain), 2nd Yield Strength (Y2 Strength) and 2nd Yield Strain (Y2 Strain) (for more details see SI). The gamma- and X-ray-irradiated PE/EVOH/PE films tested in the MD have a similar UTS to the non-sterile film (~17 MPa). Regarding the PE/EVOH/PE film tested in the TD (Figure 3b), the gamma and X-ray irradiations seem to have a lower UTS (~16 MPa) compared to the non-sterile film (~17.5 MPa). The tensile curves follow the same tendency regardless of the irradiation technology. They show a double yield characteristic of LDPE [33] in engineering stress–strain curves, and no yield shift is observed. This gives us the indication that crystallites are not impacted, regardless of the irradiation technology used. The yield stress seems to be higher for the electron-beam-irradiated sample than for the other irradiation technologies, probably due to the use of a different lot. The UTS obtained for the electron-beam-irradiated sample is also slightly higher (~19 MPa). All observations are in accordance with results observed in other studies [44,45].

The UTS data were further compared between irradiation technologies (gamma vs. X-rays, gamma vs. E-beam, X-rays vs. E-beam) with a *t*-test. All *p*-values*_t_*_-test_ are reported in the Appendix A. Regarding the UTS results obtained for the MD samples, the *p*-values*_t_*_-test_ are <0.05, and we cannot draw conclusions on the equivalency with this current dataset. The *t*-test does not consider the variabilities of the methodology, as reported during our reproducibility study (Table 2) and used to draw the UTS plots (Figure 3). For the UTS results obtained for the TD samples, only the difference between the non-sterile vs. electron-beam-irradiated samples is not significantly different.

Hypothesis tests (TOST) were also performed (*p*-values_TOST_ are in Appendix A), and for all conditions, *p*-values_TOST_ are <0.05. Equivalences were observed for UTS in the MD and TD.

**Figure 3 polymers-15-02799-f003:**
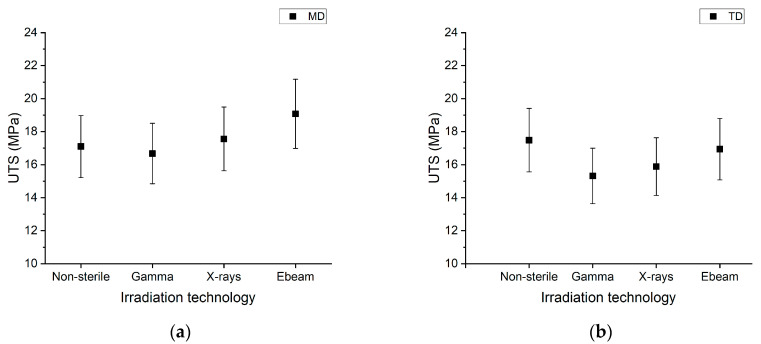
Ultimate tensile strength (UTS) for MD (**a**) and for TD (**b**) as a function of the irradiation technology. The mean is plotted with a variability (±3δ) defined in the reproducibility study outputs in Table 2. Average data of two batches. Raw data are given in the Appendix A.

Figure 4 displays the elongation at break of the PE/EVOH/PE film irradiated at 50 kGy under three irradiation technologies. The elongation at break for the film tested in the MD (Figure 4a) is obtained at ~420%, regardless of the irradiation technology. There is no difference between the non-sterile film and the irradiated films. For the film tested in the TD (Figure 4b), the elongation at break is ~540% for the non-sterile, whereas for the irradiated films, it is obtained at ~480% for gamma, X-ray and electron beam irradiation. No difference is observed regarding the stress hardening position (mainly observed ~250% in the TD and present for each irradiation technology). It can indicate that the cross-linking of the PE and the modification of EVOH occur in the same proportion in all radiation process conditions (dose and sources). This also indicates that the whole film keeps the same flexibility, regardless of the irradiation technology.

All the TOST equivalence tests performed in parallel show no impact of the irradiation technologies on the elongation at break (MD or TD). All *p*-values_TOST_ are recorded in the Appendix A. The *t*-tests performed on the MD elongation results reveal that non-sterile vs. X-ray, gamma vs. X-ray irradiation and electron beam vs. X-ray irradiation are statistically significant different. The *t*-tests results on TD elongation show that only X-ray vs. gamma irradiation results are statistically non-significantly different. All *p*-values*_t_*_-test_ are recorded Appendix A.

Greene et al. reports that an increase in the tensile strength is characterized by a higher molecular weight, whereas a decrease in the tensile strength is characterized by a broader molecular weight distribution [46]. The similar tensile strength obtained with the three irradiation technologies indicates that the molecular weight is modified in an equivalent proportion, regardless of the irradiation technology used [46]. It also shows that the number of entanglements in the polymer remains at an equivalent amount in the assessed materials with the three irradiation technologies. Bond cleavage and cross-linking occurring upon irradiation do not impact the overall mechanical properties of the multilayer film. This conclusion agrees with a previous study [33].

**Figure 4 polymers-15-02799-f004:**
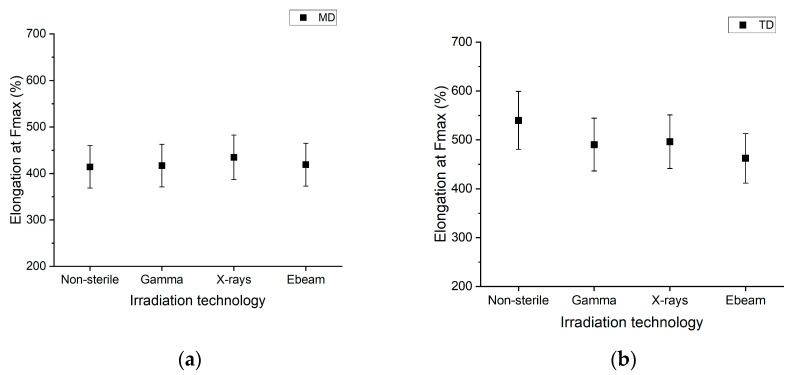
Elongation for MD (**a**) and for TD (**b**) as a function of the irradiation technology. The mean is plotted with a variability (±3δ) defined in the reproducibility study outputs in Table 2 and relies on two batches and from six to eight specimens/batch. Raw data are given in the Appendix A.

### 3.2. Thermal Relaxations

DMA (Dynamic mechanical analysis) was performed on the PE/EVOH/PE multilayer film to investigate the thermal and viscoelastic properties. Figure 5 shows three curves: the storage modulus E′ (triangles), the loss modulus E″ (squares) and tan δ (circles) in the machine direction (MD) without irradiation (non-sterile, black) and after gamma, X-ray and electron beam irradiations (red, blue and pink curves, respectively).

The storage modulus E’ is proportional to the stored energy, the loss modulus is proportional to the energy irreversibly converted to heat and tan delta is the loss factor. The maximum values of the loss modulus and tan δ correspond to the value of the β transition and γ transition for the non-sterile and sterile samples.

Polyethylene is well known to have three relaxation mechanisms, referred to as α, β and γ [47,48,49]. α relaxation is ascribed to chain motion in the crystalline phase, whereas the β and γ relaxations are related to chain motion in the amorphous phase. The γ relaxation, in some studies, is defined as the glass transition temperature [47], while in other studies, the glass transition in polyethylene and polyethylene/α-olefin-copolymers is the β and not the γ transition [48].

All phenomena (such as cross-linking, chain scission, etc.) happened to an extent that did not change the motion in the molecular chain segment, as observed per DMA.

The glass transition (γ transition) and β transition temperatures are displayed in Figure 6. A similar temperature is obtained for the γ transition (for TD and MD samples; Figure 6a and 6b, respectively), regardless of the irradiation technology. Regarding the β transition (for TD and MD samples; Figure 6c and 6d, respectively), the temperature for the electron-beam-irradiated samples is slightly lower than the non-sterile, gamma- and X-ray-irradiated samples, surely due to curve processing variations.

The *t*-tests performed on MD-γ transition results revealed that non-sterile vs. E-beam irradiation and gamma vs. X-ray irradiations are statistically non-different. The *t*-test results on TD-γ transition data show that only X-ray vs. E-beam irradiation results are statistically significantly different. All *p*-values*_t_*_-test_ are recorded Appendix A.

The *t*-tests performed on MD-β transition results revealed that E-beam vs. non-sterile, E-beam vs. gamma irradiations and E-beam vs. X-ray irradiations are statistically significantly different. The *t*-test results on the TD-β transition show that the E-beam vs. non-sterile and E-beam vs. gamma irradiations results are statistically significantly different. All *p*-values*_t_*_-test_ are recorded Appendix A.

Hypothesis tests (TOST) were also performed (*p*-values_TOST_ are in Appendix A–d), and for all conditions, *p*-values_TOST_ are <0.05. Equivalences are observed for γ and β transition temperatures (MD or TD).

There is no shift in the relaxation temperatures in irradiated samples (within methodology uncertainties), meaning that the chain motions in the amorphous phase (related to β and γ transitions) are equivalent under gamma and X-ray irradiations. No further crystallization processes occur during irradiation with the different radiation sources.

Polyethylene is a semicrystalline synthetic polymer containing some regions with order (arrangement of the polymer chains) and some other disordered amorphous regions. The manufacturing process creates preferential polymer chain orientation in thin films. There is no influence of the irradiation technology in the crystalline and amorphous parts of the polymers after the film manufacturing process. We showed no change in the thermal and viscoelastic properties for the PE layer, denoting also the absence of significant alteration in structuration (molecular size and cross-linking).

## 4. Conclusions

Variations which may have an impact on the quality or performance of a biopharmaceutical film, such as changes in the manufacturing process, e.g., the sterilization technology, were assessed after irradiation. In this paper, tensile (with UTS and elongation at break parameters) and DMA tests were used to monitor mechanical, thermal and viscoelastic properties. It has been shown that no difference is induced, regardless of the irradiation technology used on the material at 50 kGy (using TOST method). These observations are in good agreement with one of our previous studies, which showed no modification in thermal relaxations recorded by DSC [50]. Gamma, electron beam and X-ray irradiation have the same impact on PE/EVOH/PE multilayer film mechanical properties, even for low temperatures, meaning that X-ray and electron beam irradiations could be possible for freeze and thaw applications.

The comparison between gamma, E-beam or X-ray irradiations on the mechanical property effects of PE/EVOH/PE multilayer film have been studied in this paper. To compare and conclude on any potential difference regarding the impact of the irradiation technologies, two statistical methods were used: the two-sample *t*-test and the TOST. We showed that these can afford different conclusions regarding the impact of irradiation technologies on mechanical properties. We observed that the EACs are dependent on sample size, sampling homogeneity and standard deviation of the reference group(s). They can become very high for small sample sizes and large standard deviations. The equivalence criterion may not be strict enough in this case. Whatever the statistical method, the differences observed are small between each irradiation technology and rather reflect the variation in sampling and testing methodology.

## Figures and Tables

**Figure 1 polymers-15-02799-f001:**
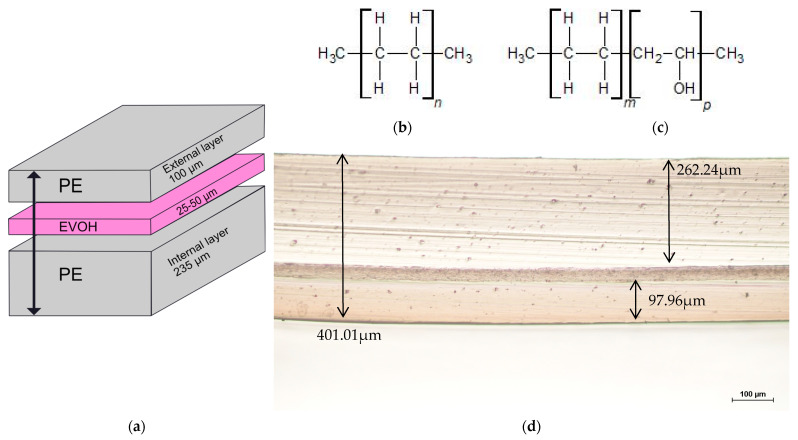
(**a**) Representation of PE/EVOH/PE multilayer film, (**b**) chemical structure of PE, (**c**) chemical structure of EVOH, (**d**) image of cross-section of multilayer film.

**Figure 2 polymers-15-02799-f002:**
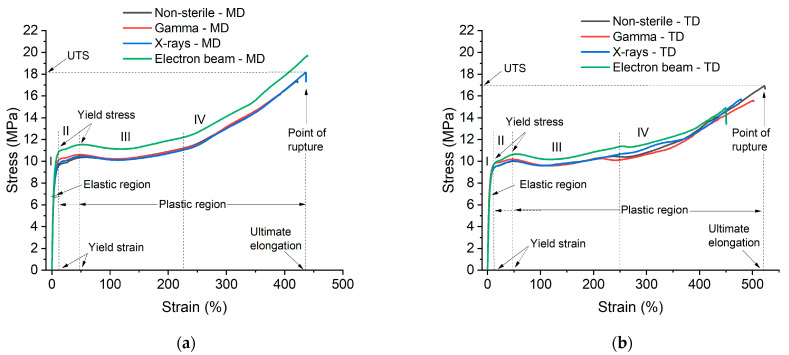
Tensile curves of PE/EVOH/PE in MD (**a**) and TD (**b**) (one batch, one specimen for each direction)—non-sterile in black, gamma-irradiated sample in red, X-ray-irradiated sample in blue, electron-beam-irradiated sample in green. A dose of 50 ± 5 KGy was used for all technologies. UTS and ultimate elongation values are shown in Figure 3 and Figure 4.

**Figure 5 polymers-15-02799-f005:**
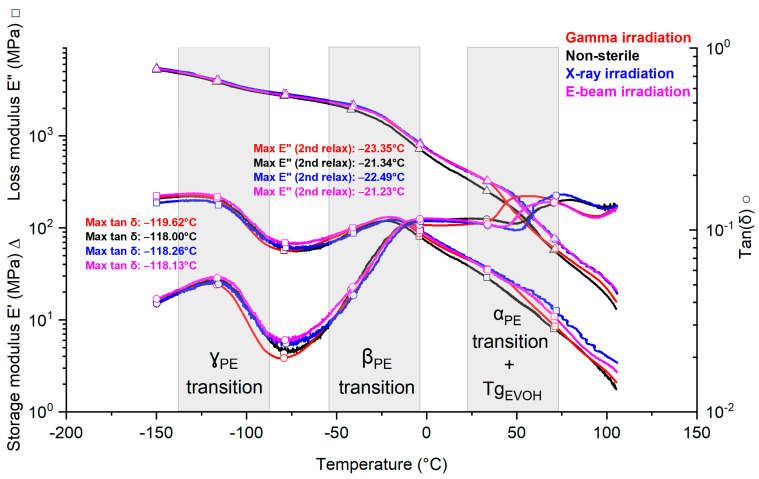
Example of DMA curves of PE/EVOH/PE in MD (one batch, one specimen). ∆ for storage modulus E′, □ for loss modulus E″ and ○ for tan δ. Gamma irradiation curves are red, non-sterile curves are black, X-ray curves are blue and electron beam curves are pink. TD curves show similar trends between gamma- and X-ray-irradiated samples and are available in the Figure 2.

**Figure 6 polymers-15-02799-f006:**
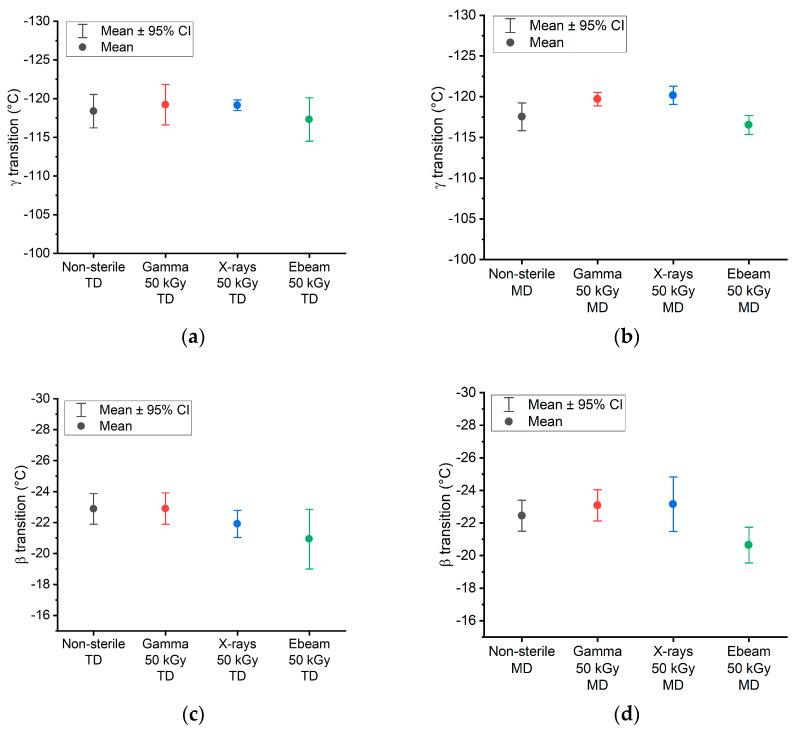
γ transition TD (**a**), γ transition MD (**b**), β transition TD (**c**) and β transition MD (**d**) for different irradiation technologies. The error bars were calculated based on two batches (except for E-beam) and from two to five specimens/batch. Raw data are given in the Appendix A.

**Table 1 polymers-15-02799-t001:** Mechanical properties investigated: tensile test and dynamical mechanical analysis (DMA) for different batches, irradiation doses (kGy) and ageing between irradiation and mechanical tests.

	Tensile Test			DMA		
	Batch #	Dose (kGy)	Ageing (Months)	Batch #	Dose (kGy)	Ageing (Months)
Gamma	Batch 1	54.3 ± 2.7	2	Batch 3	55.2 ± 2.8	6
	Batch 2	48.2 ± 2.1	2	Batch 2	48.2 ± 2.1	4
X-rays	Batch 3	55.2 ± 2.8	2	Batch 6	50.8 ± 1.4	6
	Batch 2	54.7 ± 2.5	2	Batch 2	53.1 ± 1.7	4
E-beam	Batch 4	52.0 ± 2.6	2	Batch 7	54.1 ± 9.5	12
	Batch 5	54.8 ± 2.8	2	n.a.	n.a.	n.a.
Non-sterile	Batch 2	0	n.a. ^a^	Batch 7	0	n.a.

^a^ n.a.: not applicable.

**Table 2 polymers-15-02799-t002:** Summary of tensile strength reproducibility study outputs performed on PE/EVOH/PE multilayer film and defined δ (EAC).

	UTS (Ultimate Tensile Stress)	Elongation
Six-sigma variation	22%	21%
Criteria δ	35%	25%

## Data Availability

Research data are provided in Appendix A.

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
