# Peer review of "Gamma, E-Beam and X-ray Irradiations on PE/EVOH/PE Multilayer Film: An Industrial Point of View Regarding the Impact on Mechanical Properties"

_polymers, 2023, doi:10.3390/polym15132799_

Round 1

Reviewer 1 Report

            In the article “Gamma, E-beam and X-ray irradiations on PE/EVOH/PE multi- 2 layer film: an industrial point of view regarding the impact on 3 mechanical properties” the mechanical properties of a PE/EVOH/PE film were studied using tensile tests and dynamical mechanical analysis after each irradiation technology (i.e. gamma, electron beam and X-ray irradiations).          

·                     In the Introduction part, the authors discuss about the applications of polymeric films (polyethylene/ethylene vinyl alcohol/polyethylene (PE/EVOH/PE) multilayer film) and the necessity to sterilize them when they are used in the pharmaceutical and/or food area. It is mentioned that the most common method of sterilization is the gamma-radiation but also exist two other alternative methods such as electron beam irradiation and X-ray irradiation.

            In this point, I consider that a short phrase should be introduced which describes the limitations and advantages of each individual method on the sterilization of polymeric materials.

·                     The purpose of the present article is well underlined to be “the impact of three irradiation technologies, X-ray, E-beam and gamma irradiations is investigated in industrial conditions at a single dose of 50 kGy on single use bags made up of PE/EVOH/PE multilayer film”.

            What does “industrial conditions” refer to? I think they should at least be described in relation to laboratory conditions (I suppose).

            To connoisseurs, the “single dose of 50 kGy” expression may sound like a lot, but to a general reader, this value must be explained in terms of what kind of measurement? Why this dose? Why one and not several?

·                     The novelty of this article (as I understand it) refers to the fact that although there are tests on the modification of physico-chemical properties of monolayer polymer films by irradiation with all three techniques, there is not much data and especially not a clear comparison between methods on multilayer polymer films. This should be more clearly pointed out in the end of the introductory part, so that the reader understands what is new in this article compared to all other citations.

·                     What is the novelty degree of the presented article compared to your recent article

[27]  - Girard-Perier, N.; Claeys-Bruno, M.; Marque, S.R.A.; Dupuy, N.; Gaston, F.; Dorey, S. Monitoring of peroxide in gamma irradiated PE/EVOH/PE multilayer film using methionine probe. Food and Bioproducts Processing 2022,132, 226–232, doi:10.1016/j.fbp.2022.02.001)

[47] Girard-Perier, N.; Claeys-Bruno, M.; Marque, S.R.A.; Dupuy, N.; Gaston, F.; Dorey, S. Effects of X-ray, electron beam and gamma irradiation on PE/EVOH/PE multilayer film properties. Chemical communications (the year of publication is missing)

            Both articles reports effects of X-ray, electron beam and gamma irradiation on PE/EVOH/PE multilayer film properties.

·                     60 percent of the cited articles are older than 10 years, some even older. I believe that in 10 years techniques evolve exponentially

·                     In figure 1 (a) thetext in the figure is very small, not readable. It should be modified to be readable. Exactly the same figure as in reference [27]. Needs some modifications.

·                     I think that the expression "Structure of PE/EVOH/PE (a), structure of PE (b) and EVOH (c)" in the caption of Fig 1 is not very conclusive. Image (a) it is not really a structure but a graphical representation of the multi-layer polymer  while for (b) and (c) we can talk about a chemical structure of PE respectively EVOH. Personally I think this caption should be revised

·                     Sections 2.1 and 2.2. are almost identical to similar sections in a previous article, reference [27]. Although it is the same method and the same research group, one should be careful that the texts are not the same.

·                     In section 2.2. Storage conditions the expression “has been performed according to internal industrial procedures” (lines 90-91) is not clear. What means “internal industrial procedures”?

·                     In section 2.3. “Irradiation technologies”, expression should be techniques, or methods of irradiation, instead of technologies.

·                     In Table 1, there are two columns “Tensile”, “DMA”, Additional explanations are needed in the caption of the Table1, regarding these two columns.

·                     In Table 1., as I understand the samples were irradiated once for Tensile measurements and once for DMA measurements. Why are they not the same Batch, for comparison? Doesn't matter? Why isn’t the same sample sets (batch) used for all three irradiation techniques? Why are the storage times different?

            For a clear comparison on the effects of the three irradiation techniques on mechanical properties, shouldn't a uniform protocol be followed, with dose, storage time, same sample set?

·                     Section 2.4. Tensile strength “, it should be renamed “Tensile strength measurement”. In

·                     Table 1 SI and 2SI, table related to the tensile strength measurement, the code of the sample sets is named differently than in Table 1 of the article (i.e. #6230665). A uniform code must be respected so that the reader can easily follow the measurements flow.

·                     The measurements are well performed and support the proposed goal of performing an evaluation of the effect of three irradiation methods on the mechanical properties (Tensile strength, Thermal relaxations) of a multilayer polymeric material only that it does not provide conclusive results except that there are no significant differences between the three techniques evaluated. There are a lot of technical terms that describe the measurements, the changes in values, but these are not compared with literature values, that suggest, that the mechanical changes supported by the multilayer polymeric material are in positive or negative direction.

·                     From the Conclusions section, it is understood that the two statistical evaluation methods (the two-sample t-test and the TOST) are not sufficient to have a full picture of the effect of the three irradiation methods on the physical properties, or so it is understood. (“We showed that it can afford different conclusions regarding the impact of irradiation technologies on mechanical properties (lines 325-326), “The equivalence criterion could be not strict enough in this case.” (line 329)

·                     “It has been shown that no difference is induced whatever the irradiation technology used on the material at 50 kGy (using TOST method).

            This conclusion has already been assumed in a previous article [47], as the authors themselves state. “These observations are in good agreement with one of our previous study published which showed no modification in thermal relaxations recorded by DSC (line 335-336)”.

·                     Has another irradiation value been tried? Or is this an accepted standard value?

·                     “Gamma, electron beam and X-rays irradiation have the same impact on PE/EVOH/PE multilayer film mechanical properties, even for low temperatures meaning that X-rays and electron beam irradiations could be possible for freeze and thaw applications (lines 337-340)”

            However, what would be the advantages and disadvantages of one method over the other? I'm sure there must be.

·                     Personally I think that the conclusions section should be re-evaluated so as to concretely underline the conclusions of the present article and the reader can concretely understand which of the three irradiation/sterilization methods is optimal for the material presented.

Author Response

Point 1: In the Introduction part, the authors discuss about the applications of polymeric films (polyethylene/ethylene vinyl alcohol/polyethylene (PE/EVOH/PE) multilayer film) and the necessity to sterilize them when they are used in the pharmaceutical and/or food area. It is mentioned that the most common method of sterilization is the gamma-radiation but also exist two other alternative methods such as electron beam irradiation and X-ray irradiation. In this point, I consider that a short phrase should be introduced which describes the limitations and advantages of each individual method on the sterilization of polymeric materials.

Response 1:  Details on methods have been added in the manuscript.

Point 2: The purpose of the present article is well underlined to be “the impact of three irradiation technologies, X-ray, E-beam and gamma irradiations is investigated in industrial conditions at a single dose of 50 kGy on single use bags made up of PE/EVOH/PE multilayer film”. What does “industrial conditions” refer to? I think they should at least be described in relation to laboratory conditions (I suppose).

Response 2: Industrial conditions correspond to dose range and to storage conditions described in 2.2.

Point 3: To connoisseurs, the “single dose of 50 kGy” expression may sound like a lot, but to a general reader, this value must be explained in terms of what kind of measurement? Why this dose? Why one and not several?

Response 3: All answers to the R1 are given in this reference.

“Dose range usually used for biopharmaceutical industries is 25–45 kGy (ISO 11137-1International Organization for Standardization, Sterilization of HealthCare Products - Radiation - Part 1: Requirements for Development,Validation and Routine Control of a Sterilization Process for MedicalDevices, 2006).”

Point 4: The novelty of this article (as I understand it) refers to the fact that although there are tests on the modification of physico-chemical properties of monolayer polymer films by irradiation with all three techniques, there is not much data and especially not a clear comparison between methods on multilayer polymer films. This should be more clearly pointed out in the end of the introductory part, so that the reader understands what is new in this article compared to all other citations.

    What is the novelty degree of the presented article compared to your recent article

[27]  - Girard-Perier, N.; Claeys-Bruno, M.; Marque, S.R.A.; Dupuy, N.; Gaston, F.; Dorey, S. Monitoring of peroxide in gamma irradiated PE/EVOH/PE multilayer film using methionine probe. Food and Bioproducts Processing 2022,132, 226–232, doi:10.1016/j.fbp.2022.02.001) SMASH

[47] Girard-Perier, N.; Claeys-Bruno, M.; Marque, S.R.A.; Dupuy, N.; Gaston, F.; Dorey, S. Effects of X-ray, electron beam and gamma irradiation on PE/EVOH/PE multilayer film properties. Chemical communications (the year of publication is missing)

Response 4: In this paper we discuss only issues concerning oxidation of solution which are stored in sterilized bags. In the manuscript we are focused on mechanical properties. We precise in the introduction that the paper concern that mechanical performance.

Point 5: Both articles reports effects of X-ray, electron beam and gamma irradiation on PE/EVOH/PE multilayer film properties.

60 percent of the cited articles are older than 10 years, some even older. I believe that in 10 years techniques evolve exponentially

Response 5: We added 3 new references and removed the 3 oldest references.

Point 6: In figure 1 (a) the text in the figure is very small, not readable. It should be modified to be readable. Exactly the same figure as in reference [27]. Needs some modifications.

Response 6: We changed the figure 1(a)

Point 7: I think that the expression "Structure of PE/EVOH/PE (a), structure of PE (b) and EVOH (c)" in the caption of Fig 1 is not very conclusive. Image (a) it is not really a structure but a graphical representation of the multi-layer polymer  while for (b) and (c) we can talk about a chemical structure of PE respectively EVOH. Personally I think this caption should be revised

Response 7:  We improved the figure 1.

Point 8: Sections 2.1 and 2.2. are almost identical to similar sections in a previous article, reference [27]. Although it is the same method and the same research group, one should be careful that the texts are not the same.

Response 8: We decided to repeat that sections to re-explain the industrial conditions for sampling.

Point 9: In section 2.2. Storage conditions the expression “has been performed according to internal industrial procedures” (lines 90-91) is not clear. What means “internal industrial procedures”?

Response 9: We corrected the term “internal industrial procedures”. 

Point 10: In section 2.3. “Irradiation technologies”, expression should be techniques, or methods of irradiation, instead of technologies.

Response 10: Modification done.

Point 11: In Table 1, there are two columns “Tensile”, “DMA”, Additional explanations are needed in the caption of the Table1, regarding these two columns.

Response 11: We improved the caption.

Point 12: In Table 1., as I understand the samples were irradiated once for Tensile measurements and once for DMA measurements. Why are they not the same Batch, for comparison? Doesn't matter? Why isn’t the same sample sets (batch) used for all three irradiation techniques? Why are the storage times different?

Response 12:  We gathered data over a period of two years. It encompasses potential variabilities.

Point 13: For a clear comparison on the effects of the three irradiation techniques on mechanical properties, shouldn't a uniform protocol be followed, with dose, storage time, same sample set?

Response 13: This project evaluates the robustness of the mechanical test data measured in industrial conditions.

Point 14: Section 2.4. Tensile strength “, it should be renamed “Tensile strength measurement”. In

Response 14: Modification done

Point 15: Table 1 SI and 2SI, table related to the tensile strength measurement, the code of the sample sets is named differently than in Table 1 of the article (i.e. #6230665). A uniform code must be respected so that the reader can easily follow the measurements flow.

Response 15: Modification done

Point 16: The measurements are well performed and support the proposed goal of performing an evaluation of the effect of three irradiation methods on the mechanical properties (Tensile strength, Thermal relaxations) of a multilayer polymeric material only that it does not provide conclusive results except that there are no significant differences between the three techniques evaluated. There are a lot of technical terms that describe the measurements, the changes in values, but these are not compared with literature values, that suggest, that the mechanical changes supported by the multilayer polymeric material are in positive or negative direction.

Response 16: Reference have been added and corresponding paragraph was changed.

Point 17: From the Conclusions section, it is understood that the two statistical evaluation methods (the two-sample t-test and the TOST) are not sufficient to have a full picture of the effect of the three irradiation methods on the physical properties, or so it is understood. (“We showed that it can afford different conclusions regarding the impact of irradiation technologies on mechanical properties (lines 325-326), “The equivalence criterion could be not strict enough in this case.” (line 329)

“It has been shown that no difference is induced whatever the irradiation technology used on the material at 50 kGy (using TOST method).

This conclusion has already been assumed in a previous article [47], as the authors themselves state. “These observations are in good agreement with one of our previous study published which showed no modification in thermal relaxations recorded by DSC (line 335-336)”.

Has another irradiation value been tried? Or is this an accepted standard value?

Response 17: Indeed, the 50kGy dose is the qualification dose used for all industrial products. However, we tried higher doses for other properties, and we would certainly apply the same strategy to give further insights later on.

Point 18: “Gamma, electron beam and X-rays irradiation have the same impact on PE/EVOH/PE multilayer film mechanical properties, even for low temperatures meaning that X-rays and electron beam irradiations could be possible for freeze and thaw applications (lines 337-340)”

However, what would be the advantages and disadvantages of one method over the other? I'm sure there must be.

Response 18: There would be no differences in using whether technologies from a performance point of view. The main difference might lay on the sterilization outputs not evaluated presently.

Point 19: Personally I think that the conclusions section should be re-evaluated so as to concretely underline the conclusions of the present article and the reader can concretely understand which of the three irradiation/sterilization methods is optimal for the material presented.

Response 19: We reviewed the conclusion plan to better emphasize the impacts on materials.

Reviewer 2 Report

The paper takes up the subject of Gamma, E-beam and X-ray irradiations on PE/EVOH/PE multilayer film, from an industrial point of view regarding the impact on mechanical properties. The authors reviewed the literature dealing with this issue. In "Materials and Methods" they presented PE/EVOH/PE multilayer film, storage conditions, irradiation technologies, tensile strength,  DMA and statistical analyses connected with equivalency assessment, tensile strength EAC (the equivalency acceptance criterium) and thermal properties EAC. In "Results" the authors presented, for example, tensile curves of PE/EVOH/PE in MD (machine direction) and TD (transverse direction), ultimate tensile strength for MD and TD, and elongation for MD and TD, as a function of the irradiation technology, and the example of DMA curves of PE/EVOH/PE in MD. 

The authors stated among others that the Gamma, electron beam and X-rays irradiation have the same impact on PE/EVOH/PE multilayer film mechanical properties, even for low temperatures.

Please describe the accuracy of the obtained results.

The article can be published after linguistic verification. 

Author Response

The paper takes up the subject of Gamma, E-beam and X-ray irradiations on PE/EVOH/PE multilayer film, from an industrial point of view regarding the impact on mechanical properties. The authors reviewed the literature dealing with this issue. In "Materials and Methods" they presented PE/EVOH/PE multilayer film, storage conditions, irradiation technologies, tensile strength,  DMA and statistical analyses connected with equivalency assessment, tensile strength EAC (the equivalency acceptance criterium) and thermal properties EAC. In "Results" the authors presented, for example, tensile curves of PE/EVOH/PE in MD (machine direction) and TD (transverse direction), ultimate tensile strength for MD and TD, and elongation for MD and TD, as a function of the irradiation technology, and the example of DMA curves of PE/EVOH/PE in MD. 

The authors stated among others that the Gamma, electron beam and X-rays irradiation have the same impact on PE/EVOH/PE multilayer film mechanical properties, even for low temperatures.

Please describe the accuracy of the obtained results.

The accuracy is given for example in Table 2 in summary; details are given in SI.

The article can be published after linguistic verification

Reviewer 3 Report

The comparison between gamma, E-beam or X-ray irradiations on the mechanical property effects of PE/EVOH/PE multilayer film have been studied in this paper. To compare and conclude on any potential difference regarding the impact on the irradiation technologies, the author used two statistical methods: the two-sample t- test and the TOST, which can afford different conclusions regarding the impact of irradiation technologies on mechanical properties. However, before the final publication, there are some major errors in the manuscript, which should be corrected.

1. In Figure 2, it is recommended to describe or explain the tensile curves of the PE/EVOH/PE film obtained after each irradiation technology (i.e., gamma in red, X-rays in blue and electron beam in green).

2. In Figure 4,for the film tested in TD, why the elongation at break for the non-sterile is higher than the irradiated films, Please provide an appropriate explanation for this.

3. In Figure 5,It is recommended to provide an appropriate explanation for the DMA curves of PE/EVOH/PE in MD, for example , making some appropriate description of feature points or feature intervals in the diagram.

4. In terms of format, Figures 3 and 4 should remain in the same format.

5. It is recommended to indicate at what irradiation intensity the test was conducted in Figure 2.

6. The text lacks explanations for Figures 2 and 5.

7. The text lacks explanations for Figures 2 and 5.

8. It is recommended that Figure 6 not be spread across pages.

9. If Figure 3 is based on Figure 2, it is recommended to place Figures 2 and 3 together.

10. The "Introduction" section is a bit tedious and needs to be modified to logically express the background and innovation of this work.

11. The author mentioned that this work involves the production of multi-layer films, and suggests that the author can display the multi-layer films more intuitively through physical images.

12. The images in the manuscript are too scattered. It is recommended to recombine them to improve the readability of the manuscript.

13. Table 1 has a problematic format and requires modification.

14. The table format and sorting in the supplementary information are a bit confusing and need to be modified.

15. It is recommended that the authors explain and analyze Figure 2 in order to better understand the differences in mechanical direction (MD) and transverse direction (TD) stress and strain curves.

16. It is recommended to annotate the corresponding images for better understanding of different descriptions of test results in lines 214 to 220.

17. Lines 241-242 mention that no differences is observed regarding the stress hardening position. It is recommended that the author indicate the corresponding image for easy understanding.

18. In figure 1, the table header and picture are placed on the same page; In table 1, the lines are missing. figure 5 The chart is irregular.

19. The main research object of this paper is statistical analysis, and the knowledge related to polymers is not outstanding or professional enough.

20. Lines 268-279 describes the three transitions corresponding to the movement of chain segments, but the lack of the difference between different chain segments corresponding to these three transitions and the changes in the curve caused by changes in molecular chain segments (such as crosslinking, degradation, etc.).

21. In the experiment part, the intensity of spline 2 treated with gamma rays is significantly lower than that of the control group, about 10%, which does not meet the requirements of control variable conditions.

Author Response

Point 1: In Figure 2, it is recommended to describe or explain the tensile curves of the PE/EVOH/PE film obtained after each irradiation technology (i.e., gamma in red, X-rays in blue and electron beam in green).

Response 1 : We added explanation for the figure 2

Point 2 : In Figure 4,for the film tested in TD, why the elongation at break for the non-sterile is higher than the irradiated films, Please provide an appropriate explanation for this.

Response 2 : There is no significant difference between the non-sterile and sterile samples, the error bars of the samples overlap.

Point 3: In Figure 5,It is recommended to provide an appropriate explanation for the DMA curves of PE/EVOH/PE in MD, for example , making some appropriate description of feature points or feature intervals in the diagram.

Response 3: We added explanation for the DMA curves of PE/EVOH/PE

Point 4: In terms of format, Figures 3 and 4 should remain in the same format.

Response 4: Modification done

Point 5: It is recommended to indicate at what irradiation intensity the test was conducted in Figure 2.

Response 5: Modification done

Point 6: The text lacks explanations for Figures 2 and 5.

Response 6: We added explanation

Point 8: It is recommended that Figure 6 not be spread across pages

Response 8: Modification done

Point 9: If Figure 3 is based on Figure 2, it is recommended to place Figures 2 and 3 together.

Response 9: We have chosen to separate the two figures for a better understanding of the tensile curves in figure 2. 

Point 10: The "Introduction" section is a bit tedious and needs to be modified to logically express the background and innovation of this work.

Response 10: The introduction was improved.

Point 11: The author mentioned that this work involves the production of multi-layer films and suggests that the author can display the multi-layer films more intuitively through physical images

Response 11: We added image in the figure 1.

Point 12: The images in the manuscript are too scattered. It is recommended to recombine them to improve the readability of the manuscript.

Response 12: We let the editors decide what has to be done.

Point 13: Table 1 has a problematic format and requires modification.

Response 13: Modification done.

Point 14: The table format and sorting in the supplementary information are a bit confusing and need to be modified.

Response 14: We improved the layout to ease the reading.

Point 15: It is recommended that the authors explain and analyze Figure 2 in order to better understand the differences in mechanical direction (MD) and transverse direction (TD) stress and strain curves.

Response 15: The explanation is given in 2.4.

Point 16: It is recommended to annotate the corresponding images for better understanding of different descriptions of test results in lines 214 to 220.

Response 16: Figure 2 has been annotated as requested.

Point 17: Lines 241-242 mention that no differences is observed regarding the stress hardening position. It is recommended that the author indicate the corresponding image for easy understanding.

Response 17: We improve the figure 2

Point 18: In figure 1, the table header and picture are placed on the same page; In table 1, the lines are missing. figure 5 The chart is irregular.

Response 18: We modified the figure 2

Point 19: The main research object of this paper is statistical analysis, and the knowledge related to polymers is not outstanding or professional enough.

Response 19: We added details for better understanding.

Point 20: Lines 268-279 describes the three transitions corresponding to the movement of chain segments, but the lack of the difference between different chain segments corresponding to these three transitions and the changes in the curve caused by changes in molecular chain segments (such as crosslinking, degradation, etc.).

Response 20: All phenomena (such as crosslinking, chain scission, etc) happened in extent not to change motion in molecular chain segment as observed per DMA.

Point 21: In the experiment part, the intensity of spline 2 treated with gamma rays is significantly lower than that of the control group, about 10%, which does not meet the requirements of control variable conditions.

Response 21: If refer to Figure 3 and 4, the variation observed is within the methodology uncertainty.
